# High Expression of POGK Predicts Poor Prognosis in Patients with Hepatocellular Carcinoma

Wenxiong Xu [1,2,3,†], Yanlin Huang [4,†], Yongyu Mei [1,2], Yeqiong Zhang [1,2], Qiumin Luo [1,2], Shu Zhu [1,2], Liang Peng [1,2,3], Zhiliang Gao [1,2,3], Ying Liu [1,2,3,*] and Jianguo Li [1,2,*]

1   Department of Infectious Diseases, The Third Affiliated Hospital of Sun Yat-sen University, Guangzhou 510630, China
2   Guangdong Key Laboratory of Liver Disease Research, The Third Affiliated Hospital of Sun Yat-sen University, Guangzhou 510630, China
3   Key Laboratory of Tropical Disease Control, Ministry of Education, Sun Yat-sen University, Guangzhou 510630, China
4   Department of Gastroenterology, Sun Yat-sen Memorial Hospital of Sun Yat-sen University, Guangzhou 510120, China
*   Correspondence: liuy35@mail.sysu.edu.cn (Y.L.); ljiang@mail.sysu.edu.cn (J.L.)
†   These authors contributed equally to this work.

**Abstract:** Objective: Kruppel-associated box (KRAB) proteins reportedly play a dual role in neoplastic transformation. At present, little is known about the function of the proteins encoded by the human pogo transposable element derived with KRAB domain (POGK) gene. Herein, we evaluated the prognostic significance of POGK expression in patients with hepatocellular carcinoma (HCC). Methods: The data of HCC patients was downloaded from The Cancer Genome Atlas (TCGA) database. To determine the relationship between POGK and clinical features, logistic regression was applied. Cox regression and Kaplan-Meier analyses were used to evaluate the correlation between POGK and survival rates. Gene ontology (GO) analysis and Gene set enrichment analysis (GSEA) were conducted to identify the enriched pathways and functions associated with POGK. Results: A total of 374 HCC patients were identified in TCGA. POGK was significantly upregulated in HCC and correlated with tumor status ($p = 0.036$), race ($p = 0.025$), weight ($p = 0.002$), body mass index ($p = 0.033$), histologic grade ($p < 0.001$), and alpha-fetoprotein ($p < 0.001$). High POGK expression in HCC patients correlated with a poor outcome in terms of overall survival ($p = 0.0018$), progression-free survival ($p = 0.0087$), relapse-free survival ($p = 0.045$), and disease-specific survival ($p = 0.014$), according to Kaplan-Meier analysis. Receiver operating characteristic curve analysis showed that the area under the curve of POGK expression for HCC diagnosis was 0.891. GSEA showed that high POGK expression might activate mitotic prometaphase, kinesins, homologous DNA pairing and strand exchange, MET activates PTK2 signaling pathway, G1 to S cell cycle control, Aurora B pathway, ncRNAs involved in WNT signaling pathway, hepatitis C, and ncRNAs involved in the STAT3 signaling pathway. POGK expression correlated with the abundance of adaptive and innate immunocytes in HCC. Conclusion: High expression of POGK has high diagnostic and prognostic values in patients with HCC. Moreover, POGK expression is correlated with immune infiltration in HCC.

**Keywords:** Kruppel-associated box; pogo transposable element derived with KRAB domain; hepatocellular carcinoma; prognosis

## 1. Introduction

In 2018, liver cancer was reportedly the fourth most prevalent cancer worldwide and the sixth most frequently diagnosed malignancy [1]. The 5-year relative survival rate of liver cancer was about 20%, the second lowest for cancers in the United States of America [2]. It has been established that most primary liver cancers are hepatocellular carcinoma (HCC)

which accounts for nearly 90% of cases [3]. The two primary carcinogenic infectious agents for HCC are hepatitis B virus (HBV) and hepatitis C virus (HCV) [4]. At present, there are 20 to 30 million people diagnosed with chronic hepatitis B (CHB) in China [5], and it is widely acknowledged that most liver cancers are caused by chronic HBV infection in China [6]. These findings underscore the need to discover novel biomarkers that can be used to assist clinicians during the diagnostic workup and as therapeutic targets in this patient population.

An increasing body of evidence suggests that factors involved in HCC development and progression include the pro-carcinogenic effect of hepatitis virus (such as HBV and HCV), inactivation of multiple tumor suppressor genes (such as p53), abnormal activation of oncogenes (such as K-ras), dysregulation of epigenetic events (such as microRNAs), homeobox genes, multiple signaling pathways (PI3K, MAPK, JAK/STAT, NF-κB, Wnt/β-catenin, etc.), exosomes, and the immunological liver microenvironment [7,8]. Kruppel-associated box (KRAB) proteins, including KRAB zinc finger proteins (KRAB-ZFPs or ZNFs) and KRAB-associated protein 1 (KAP1), have been associated with different aspects of human adaptive immune cell differentiation and function [9]. Interestingly, the KRAB-ZFP/KAP1 system participates in chromatin relaxation and recruitment of the DNA repairing complex [10,11], playing a dual role in cancer development. Moreover, it has been shown that KRAB-ZFP mediates the DNA damage response and the expression of oncogene and/or oncosuppressor genes [12,13]. Indeed, KAP1 regulates the p21 and p53 pathways differently [14,15]. The Pogo transposable element derived with the KRAB domain (POGK) gene is well-established to be conserved in humans and other vertebrates like the chimpanzee, rhesus monkey, dog, cow, mouse, etc. Nonetheless, the functions of proteins encoded by the human POGK gene remain unclear. The KRAB domain found at the N-terminus is involved in protein-protein interactions, and a transposase domain is present at the C-terminus, suggesting that it could potentially belong to the family of DNA-mediated transposons in humans.

Few studies have hitherto explored POGK expression and its potential prognostic impact on HCC. Herein, we aimed to analyze the importance of POGK in patients with HCC systematically. Moreover, we further analyzed the correlation between POGK and immune infiltration in the tumor microenvironment.

## 2. Materials and Methods

### 2.1. POGK Expression Validation

Oncomine (https://www.oncomine.org/resource/main.html (accessed on 3 September 2021)), Tumor Immune Estimation Resource (TIMER; cistrome.shinyapps.io/timer (accessed on 3 September 2021)) and The Cancer Genome Atlas (TCGA; https://cancergenome.nih.gov (accessed on 3 September 2021)) databases were used to validate the expression level of POGK in patients with HCC and normal subjects. Boxplots and scatter plots were used to evaluate the expression of the POGK gene in patients with HCC. POGK expression higher or lower than the median value was defined as POGK-high or POGK-low, respectively.

To validate POGK over-expression in HCC, tissue microarrays (TMAs) were used to analyze POGK expression from clinical samples of 30 HCC patients of The Third Affiliated Hospital of Sun Yat-sen University. Primary HCC tissues and adjacent normal tissues were collected from eight patients. Immunohistochemical (IHC) staining was performed on 5-μm sections of the TMAs to assess the cytoplasmic expression of POGK. TMA slides were scanned using the Aperio slide scanner and analyzed using Image Scope software (Version 12.4.0.5043) (Aperio, Leica Biosystems Inc., Buffalo Grove, IL, USA). After fixation in formalin and embedding in paraffin, two observers blinded to the histopathological features and clinical data evaluated the degree of immune staining. The immunohistochemical score was obtained based on the proportion of positively-stained tumor cells and the staining intensity. Scores rated by the two independent investigators were averaged. An optimal cut-off value was determined as follows: low expression of POGK was defined as a ratio of

(expression index score of tumor/expression index score of paired adjacent non-neoplastic tissue) < 1, and normal or high expression of POGK was defined as a ratio $\geq 1$.

### 2.2. Patient Data Source and Pprocessing

Gene expression data of patients with the corresponding clinical characteristics were downloaded from TCGA. Samples were excluded for the following reasons: (1) gene expression value was equal to zero and (2) incomplete survival information. Finally, a total of 374 patients with HCC were enrolled in this study. Data on clinical characteristics that were unavailable or unidentified were considered to be missing values.

### 2.3. Gene ontology (GO) and Kyoto Encyclopedia of Genes and Genomes (KEGG) Pathway Analysis

GO analysis, which offers the latest annotations and describes the features of genes and gene products in organisms, was used to describe biological processes (BPs), cellular components (CCs), and molecular functions (MFs) of the POGK gene. KEGG pathway analysis was used in pathway research of the POGK gene, including genetic processing, environmental processing, cellular processes, metabolism, and biological systems.

### 2.4. Gene Set Enrichment Analysis (GSEA)

GSEA was used to identify genes associated with POGK gene expression and examine the significance of differences in survival between the POGK-high or POGK-low groups. The criteria for significantly enriched gene sets included a nominal *p*-value of less than 5% and a false discovery rate of less than 25%. The relative tumor infiltration of 24 immune cell types was quantified by single-sample GSEA (ssGSEA).

### 2.5. Statistical Analysis

Logistic regression was performed to assess the association between POGK expression and clinical characteristics in HCC patients. Kaplan-Meier analysis was performed to compare the survival differences between the high and low POGK expression groups. The receiver operating characteristic (ROC) curve analysis was applied to assess the diagnostic performance of POGK. Potential prognostic factors were screened using univariate Cox analysis, while multivariate Cox analysis assessed the effect of POGK expression on survival along with other clinical characteristics. Pearson and Spearman correlation analyses were used to explore the correlation between POGK and the infiltration levels of immune cells. All statistical analyses in the present study were performed using R statistical software (version 3.5.3) or SPSS software (version 24.0). A *p*-value < 0.05 was statistically significant.

## 3. Results

### 3.1. POGK Expression Analysis

We first found that POGK was significantly upregulated in HCC via pooled comparative analysis in the Oncomine database (Figure 1A,B). Consistently, the upregulation of POGK in HCC was found in the TIMER database (Figure 1C).

Subsequently, we compared POGK expression between normal and HCC samples in TCGA database. POGK expression was significantly higher in HCC samples than in normal samples (*p* < 0.001) (Figure 1D). Moreover, POGK expression was significantly higher in HCC samples compared with paired normal samples (*p* < 0.001) (Figure 1E).

We next conducted an IHC analysis of the TMAs from 30 HCC patients for POGK expression. We observed dark brown staining in HCC tissues, suggesting POGK expression was high in HCC tissues (Figure 2). In addition, TMAs analysis revealed that POGK expression in HCC tissues was significantly higher than in adjacent normal tissues for 20 (67%) cases.

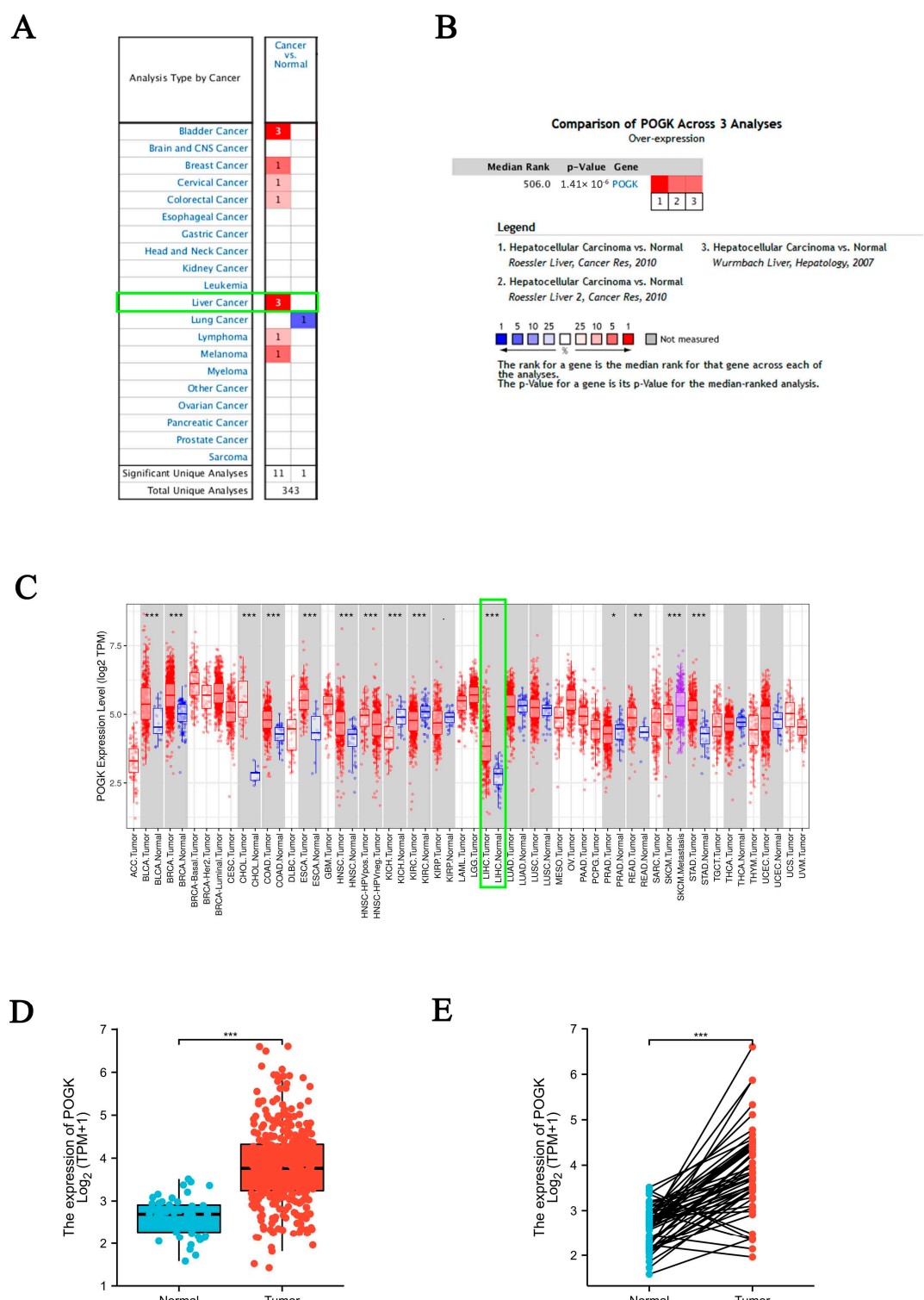

**Figure 1.** POGK expression analysis by Oncomine, TIMER, and TCGA databases. (**A**) POGK expression in different types of human cancers in Oncomine database, the numbers in the boxes represent the number of datasets, green box highlights hepatocellular carcinoma; (**B**) Upregulation (red) of POGK in hepatocellular carcinoma compared with normal tissue by Oncomine meta-analysis, the numbers in the boxes refer to the datasets below [13,14]; (**C**) POGK expression in different types of human cancers in TIMER database, green box highlights hepatocellular carcinoma; (**D**) Different POGK expression in normal and tumor tissues in TCGA database; (**E**) Different POGK expression in paired tissues in TCGA database. *: $p < 0.05$, **: $p < 0.01$, ***: $p < 0.001$.

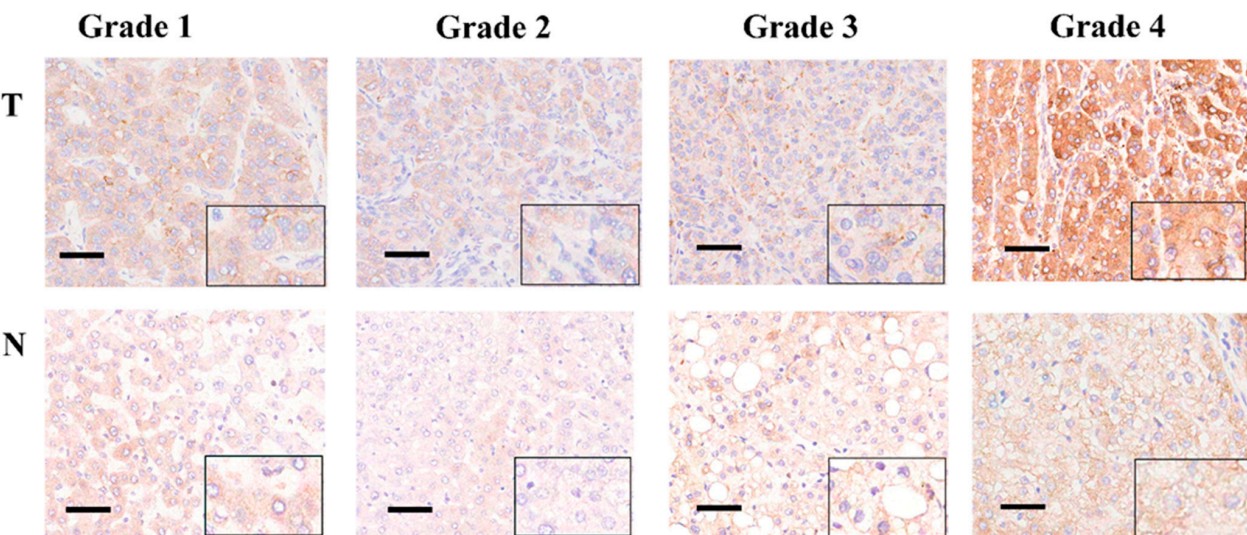

**Figure 2.** TMAs and IHC analysis of POGK expression in matched HCC tissues from patients with tumor grade 1 to 4. Top row: tumor tissues. Bottom row: adjacent normal tissues. The boxed area of each image is shown at a higher magnification in the inset. (Scale bars, 100 μm).

### 3.2. Baseline Characteristics of Patients

POGK expression data from 374 HCC patients and the corresponding clinical information were acquired from TCGA database, including patients with low expression (*n* = 187) and high expression (*n* = 187) of POGK. The detailed baseline characteristics are shown in Table 1. Among the 374 patients, 253 were male (67.6%), and 121 were female (32.4%). The pathological stage of most patients was stage I (*n* = 171, 45.7%), followed by stage II (*n* = 87, 23.3%), stage III (*n* = 85, 22.7%), and stage IV (*n* = 5, 1.3%). The tumor status included 202 patients with tumors (54.0%) and 153 tumor-free patients (40.9%). The OS event included 287 patients (76.7%) alive and 79 patients (21.1%) dead.

**Table 1.** Baseline characteristics.

| Characteristic | Low POGK Expression | High POGK Expression | *p* Value |
|:---:|:---:|:---:|:---:|
| *n* | 187 | 187 | |
| T stage, *n* (%) | | | 0.341 |
| T1 | 99 (26.7%) | 84 (22.6%) | |
| T2 | 45 (12.1%) | 50 (13.5%) | |
| T3 | 34 (9.2%) | 46 (12.4%) | |
| T4 | 6 (1.6%) | 7 (1.9%) | |
| N stage, *n* (%) | | | 0.625 |
| N0 | 120 (46.5%) | 134 (51.9%) | |
| N1 | 1 (0.4%) | 3 (1.2%) | |
| M stage, *n* (%) | | | 0.361 |
| M0 | 130 (47.8%) | 138 (50.7%) | |
| M1 | 3 (1.1%) | 1 (0.4%) | |
| Pathologic stage, *n* (%) | | | 0.132 |
| Stage I | 94 (26.9%) | 79 (22.6%) | |
| Stage II | 44 (12.6%) | 43 (12.3%) | |
| Stage III | 35 (10%) | 50 (14.3%) | |
| Stage IV | 4 (1.1%) | 1 (0.3%) | |
| Tumor status, *n* (%) | | | 0.036 * |
| Tumor free | 111 (31.3%) | 91 (25.6%) | |
| With tumor | 66 (18.6%) | 87 (24.5%) | |
| Gender, *n* (%) | | | 0.377 |

**Table 1.** *Cont.*

| Characteristic | Low POGK Expression | High POGK Expression | *p* Value |
|---|---|---|---|
| Female | 56 (15%) | 65 (17.4%) | |
| Male | 131 (35%) | 122 (32.6%) | |
| Race, *n* (%) | | | 0.025 * |
| Asian | 65 (18%) | 95 (26.2%) | |
| Black or African American | 9 (2.5%) | 8 (2.2%) | |
| White | 102 (28.2%) | 83 (22.9%) | |
| Age, *n* (%) | | | 0.133 |
| ≤60 | 81 (21.7%) | 96 (25.7%) | |
| >60 | 106 (28.4%) | 90 (24.1%) | |
| Weight, *n* (%) | | | 0.002 ** |
| ≤70 | 78 (22.5%) | 106 (30.6%) | |
| >70 | 97 (28%) | 65 (18.8%) | |
| Height, *n* (%) | | | 0.678 |
| <170 | 99 (29%) | 102 (29.9%) | |
| ≥170 | 73 (21.4%) | 67 (19.6%) | |
| BMI, *n* (%) | | | 0.033 * |
| ≤25 | 79 (23.4%) | 98 (29.1%) | |
| >25 | 91 (27%) | 69 (20.5%) | |
| Residual tumor, *n* (%) | | | 0.083 |
| R0 | 170 (49.3%) | 157 (45.5%) | |
| R1 | 5 (1.4%) | 12 (3.5%) | |
| R2 | 1 (0.3%) | 0 (0%) | |
| Histologic grade, *n* (%) | | | <0.001 *** |
| G1 | 39 (10.6%) | 16 (4.3%) | |
| G2 | 102 (27.6%) | 76 (20.6%) | |
| G3 | 39 (10.6%) | 85 (23%) | |
| G4 | 5 (1.4%) | 7 (1.9%) | |
| Adjacent hepatic tissue inflammation, *n* (%) | | | 0.092 |
| None | 69 (29.1%) | 49 (20.7%) | |
| Mild | 45 (19%) | 56 (23.6%) | |
| Severe | 11 (4.6%) | 7 (3%) | |
| AFP (ng/mL), *n* (%) | | | <0.001 *** |
| ≤400 | 123 (43.9%) | 92 (32.9%) | |
| >400 | 21 (7.5%) | 44 (15.7%) | |
| Albumin (g/dL), *n* (%) | | | 0.947 |
| <3.5 | 38 (12.7%) | 31 (10.3%) | |
| ≥3.5 | 124 (41.3%) | 107 (35.7%) | |
| Prothrombin time, *n* (%) | | | 0.145 |
| ≤4 | 103 (34.7%) | 105 (35.4%) | |
| >4 | 53 (17.8%) | 36 (12.1%) | |
| Child-Pugh grade, *n* (%) | | | 0.902 |
| A | 121 (50.2%) | 98 (40.7%) | |
| B | 11 (4.6%) | 10 (4.1%) | |
| C | 1 (0.4%) | 0 (0%) | |
| Fibrosis ishak score, *n* (%) | | | 0.329 |
| 0 | 46 (21.4%) | 29 (13.5%) | |
| 1/2 | 15 (7%) | 16 (7.4%) | |
| 3/4 | 12 (5.6%) | 16 (7.4%) | |
| 5/6 | 45 (20.9%) | 36 (16.7%) | |
| Vascular invasion, *n* (%) | | | 0.245 |
| No | 114 (35.8%) | 94 (29.6%) | |
| Yes | 52 (16.4%) | 58 (18.2%) | |
| OS event, *n* (%) | | | 0.158 |
| Alive | 129 (34.5%) | 115 (30.7%) | |
| Dead | 58 (15.5%) | 72 (19.3%) | |
| DSS event, *n* (%) | | | 0.414 |

**Table 1.** *Cont.*

| Characteristic | Low POGK Expression | High POGK Expression | *p* Value |
|---|---|---|---|
| Alive | 148 (40.4%) | 139 (38%) | |
| Dead | 36 (9.8%) | 43 (11.7%) | |
| PFI event, *n* (%) | | | 0.148 |
| Alive | 103 (27.5%) | 88 (23.5%) | |
| Dead | 84 (22.5%) | 99 (26.5%) | |
| Age, median (IQR) | 63 (53.5, 69) | 60 (51, 68) | 0.091 |

*: $p < 0.05$, **: $p < 0.01$, ***: $p < 0.001$.

### 3.3. Correlation between POGK Expression and Clinical Characteristics

As shown in Table 1 and Figure 3A–I, high expression of POGK significantly correlated with tumor status ($p = 0.036$), race ($p = 0.025$), weight ($p = 0.002$), body mass index (BMI, $p = 0.033$), histologic grade ($p < 0.001$), and alpha-fetoprotein (AFP, $p < 0.001$). Univariate analysis using logistic regression demonstrated that POGK expression was associated with clinicopathologic characteristics, typically associated with tumor aggressiveness (Table 2). Moreover, it was found that high POGK expression was significantly associated with tumor status (with tumor vs. Tumor free: odds ratio [OR] = 1.608, 95% confidence interval [CI] = 1.055–2.461, $p = 0.028$), race (White vs. Asian and Black or African American, OR = 0.585, 95% CI = 0.385–0.885, $p = 0.011$), weight (>70 kg vs. ≤70 kg, OR = 0.493, 95% CI = 0.320–0.756, $p = 0.001$), BMI (>25 vs. ≤25, OR = 0.611, 95% CI = 0.396–0.939, $p = 0.025$), histologic grade (G3 and G4 vs. G1 and G2, OR = 3.205, 95% CI = 2.064–5.033, $p < 0.001$), AFP (>400 ng/mL vs. ≤400 ng/mL, OR =2.801, 95% CI = 1.576–5.110, $p < 0.001$).

**Table 2.** Logistics regression analysis for POGK gene expression.

| Characteristics | Total (*N*) | Odds Ratio (OR) | *p* Value |
|---|---|---|---|
| T stage (T2 and T3 and T4 vs. T1) | 371 | 1.428 (0.950–2.153) | 0.087 |
| N stage (N1 vs. N0) | 258 | 2.687 (0.339–54.709) | 0.395 |
| M stage (M1 vs. M0) | 272 | 0.314 (0.015–2.487) | 0.318 |
| Pathologic stage (Stage II, Stage III, and Stage IV vs. Stage I) | 350 | 1.348 (0.886–2.055) | 0.164 |
| Tumor status (With tumor vs. Tumor free) | 355 | 1.608 (1.055–2.461) | 0.028 * |
| Gender (Female vs. Male) | 374 | 1.246 (0.808–1.927) | 0.320 |
| Age (>60 vs. ≤60) | 373 | 0.716 (0.476–1.076) | 0.109 |
| Race (White vs. Asian and Black or African American) | 362 | 0.585 (0.385–0.885) | 0.011 * |
| Weight (>70 vs. ≤70) | 346 | 0.493 (0.320–0.756) | 0.001 ** |
| Height (≥170 vs. <170) | 341 | 0.891 (0.578–1.372) | 0.600 |
| BMI (>25 vs. ≤25) | 337 | 0.611 (0.396–0.939) | 0.025 * |
| Residual tumor (R1 and R2 vs. R0) | 345 | 2.166 (0.820–6.348) | 0.131 |
| Histologic grade (G3 and G4 vs. G1 and G2) | 369 | 3.205 (2.064–5.033) | <0.001 *** |
| Adjacent hepatic tissue inflammation (Severe and Mild vs. None) | 237 | 1.584 (0.950–2.656) | 0.079 |
| AFP (ng/mL) (>400 vs. ≤400) | 280 | 2.801 (1.576–5.110) | <0.001 *** |
| Albumin (g/dL) (≥3.5 vs. <3.5) | 300 | 1.058 (0.617–1.824) | 0.839 |
| Prothrombin time (>4 vs. ≤4) | 297 | 0.666 (0.401–1.099) | 0.114 |
| Child-Pugh grade (B and C vs. A) | 241 | 1.029 (0.418–2.484) | 0.949 |
| Fibrosis ishak score (3/4 and 5/6 vs. 0 and 1/2) | 215 | 1.237 (0.722–2.123) | 0.439 |
| Vascular invasion (Yes vs. No) | 318 | 1.353 (0.852–2.154) | 0.201 |

*: $p < 0.05$, **: $p < 0.01$, ***: $p < 0.001$.

### 3.4. High Expression of POGK Is a Risk Factor for Survival in HCC

Kaplan-Meier survival analysis showed that high POGK expression correlated with a poor prognosis in HCC patients in terms of overall survival (OS) ($p = 0.0018$), progression-free survival (PFS) ($p = 0.0087$), relapse-free survival (RFS) ($p = 0.045$), disease-specific survival (DSS) ($p = 0.014$) (Figure 4A–D). After stratifying based on clinical characteristics, high POGK expression was significantly associated with poor prognosis in HCC patients with pathologic stages 2 and 3 ($p = 0.027$), pathologic stages 3 and 4 ($p = 0.0028$), pathologic

stage 3 (*p* = 0.0053), T3 stage (*p* = 0.00089), histologic grade 1 (*p* = 0.00086), male gender (*p* = 0.0014), absence of history of HBV infection (*p* = 0.00073), and history of alcohol intake (*p* = 0.03) (Figure 4E–T). Univariate Cox analysis demonstrated that high POGK expression was significantly correlated with poor overall survival in HCC patients (hazard ratio [HR] = 1.582, 95% CI = 1.112–2.249, *p* = 0.011). However, multivariate Cox analysis demonstrated that high POGK expression was not significantly correlated with poor overall survival in HCC patients (HR = 1.550, 95% CI = 0.973–2.471, *p* = 0.065) (Table 3).

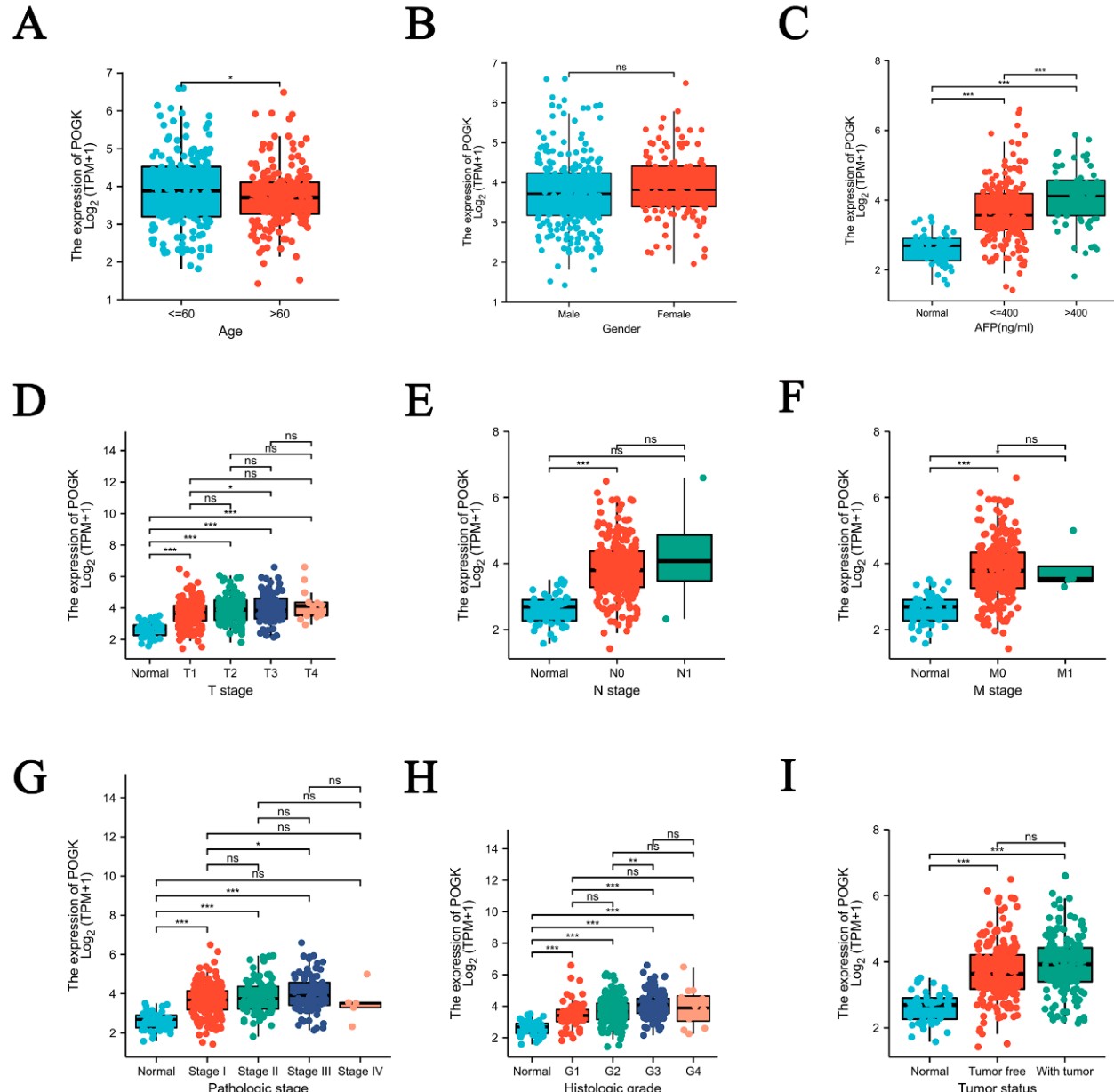

**Figure 3.** POGK expression of patients with hepatocellular carcinoma according to different clinical characteristics. (**A**) age; (**B**) gender; (**C**) AFP; (**D**) T stage; (**E**) N stage; (**F**) M stage; (**G**) pathologic stage; (**H**) histologic grade; (**I**) tumor status. ns: no significant; *: *p* < 0.05; **: *p* < 0.01; ***: *p* < 0.001.

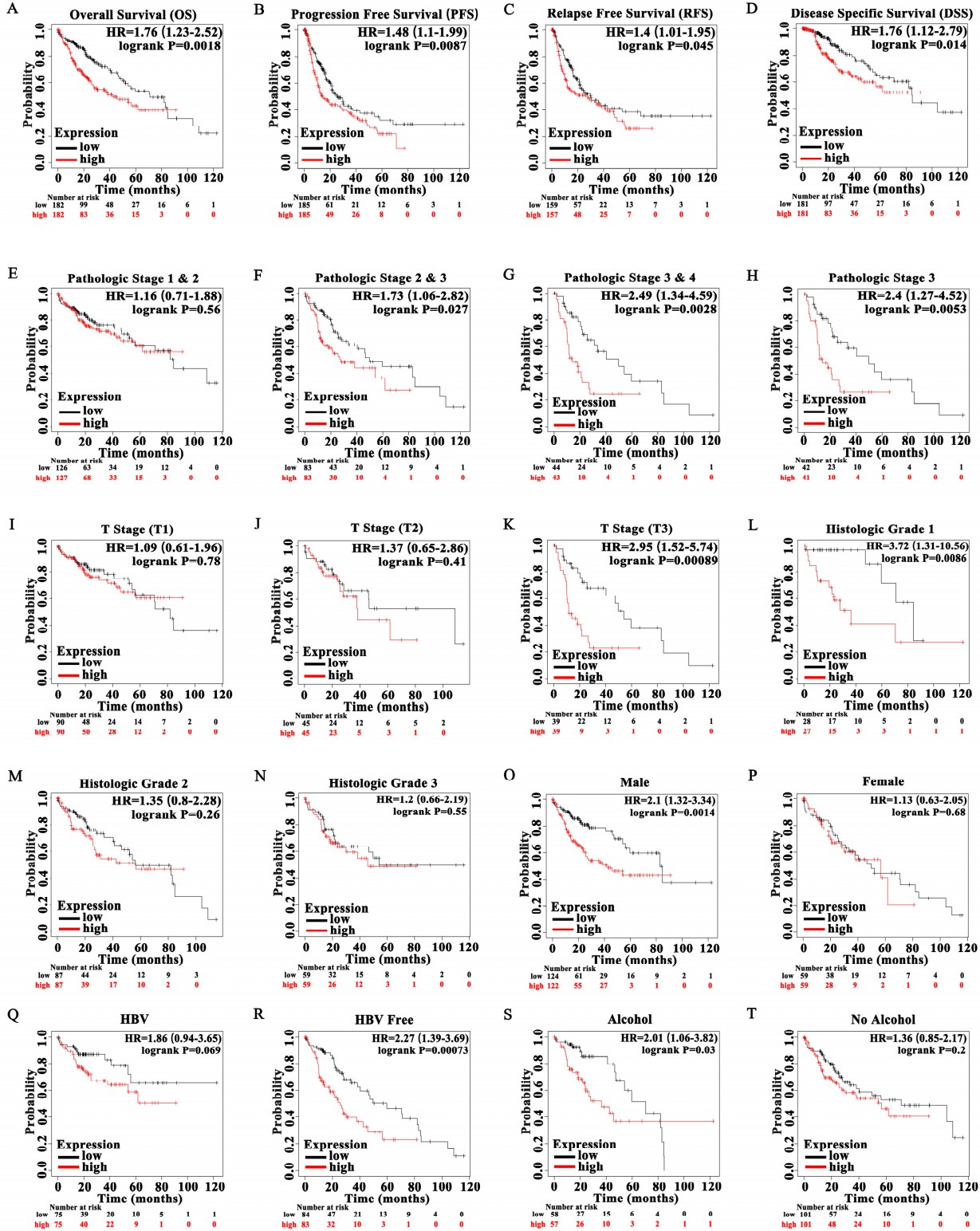

**Figure 4.** Kaplan-Meier curve for survival in patients with hepatocellular carcinoma according to high and low expression levels of POGK. (**A**) overall survival (OS); (**B**) progression-free survival (PFS); (**C**) relapse-free survival (RFS); (**D**) disease-specific survival (DSS); (**E**) pathologic stage 1 and 2; (**F**) pathologic stage 2 and 3; (**G**) pathologic stage 3 and 4; (**H**) pathologic stage 3; (**I**) T1 stage; (**J**) T2 stage; (**K**) T3 stage; (**L**) histologic grade 1; (**M**) histologic grade 2; (**N**) histologic grade 3; (**O**) male; (**P**) female; (**Q**) with history of HBV infection; (**R**) absence of history of HBV infection; (**S**) with history of alcohol intake; (**T**) absence of alcohol intake.

**Table 3.** Cox regression analysis for POGK gene expression.

| Characteristics | Total (N) | Univariate Analysis | | Multivariate Analysis | |
|---|---|---|---|---|---|
| | | Hazard Ratio (95% CI) | *p* Value | Hazard Ratio (95% CI) | *p* Value |
| Age | 373 | | | | |
| ≤60 | 177 | Reference | | | |
| >60 | 196 | 1.205 (0.850–1.708) | 0.295 | | |
| Gender | 373 | | | | |
| Male | 253 | Reference | | | |
| Female | 121 | 1.261 (0.885–1.796) | 0.200 | | |
| BMI | 336 | | | | |
| ≤25 | 177 | Reference | | | |
| >25 | 160 | 0.798 (0.550–1.158) | 0.235 | | |
| T stage | 370 | | | | |
| T1 | 183 | Reference | | | |
| T2 and T3 and T4 | 188 | 2.126 (1.481–3.052) | <0.001 *** | 0.865 (0.118–6.362) | 0.886 |
| N stage | 258 | | | | |
| N0 | 254 | Reference | | | |
| N1 | 4 | 2.029 (0.497–8.281) | 0.324 | | |
| M stage | 272 | | | | |
| M0 | 268 | Reference | | | |
| M1 | 4 | 4.077 (1.281–12.973) | 0.017 * | 2.176 (0.508–9.328) | 0.295 |
| Pathologic stage | 349 | | | | |
| Stage I | 173 | Reference | | | |
| Stage II and Stage III and Stage IV | 177 | 2.090 (1.429–3.055) | <0.001 *** | 2.690 (0.355–20.400) | 0.338 |
| Tumor status | 354 | | | | |
| Tumor free | 202 | Reference | | | |
| With tumor | 153 | 2.317 (1.590–3.376) | <0.001 *** | 1.921 (1.203–3.066) | 0.006 ** |
| Histologic grade | 368 | | | | |
| G1 and G2 | 233 | Reference | | | |
| G3 and G4 | 136 | 1.091 (0.761–1.564) | 0.636 | | |
| Adjacent hepatic tissue inflammation | 236 | | | | |
| None | 118 | Reference | | | |
| Mild and Severe | 119 | 1.194 (0.734–1.942) | 0.475 | | |
| AFP (ng/mL) | 279 | | | | |
| ≤400 | 215 | Reference | | | |
| >400 | 65 | 1.075 (0.658–1.759) | 0.772 | | |
| Albumin (g/dL) | 299 | | | | |
| <3 5 | 69 | Reference | | | |
| ≥3 5 | 231 | 0.897 (0.549–1.464) | 0.662 | | |
| Prothrombin time | 296 | | | | |
| ≤4 | 208 | Reference | | | |
| >4 | 89 | 1.335 (0.881–2.023) | 0.174 | | |
| Child-Pugh grade | 240 | | | | |
| A | 219 | Reference | | | |
| B and C | 22 | 1.643 (0.811–3.330) | 0.168 | | |
| Fibrosis ishak score | 214 | | | | |
| 0 and 1/2 | 106 | Reference | | | |
| 3/4 and 5/6 | 109 | 0.740 (0.445–1.232) | 0.247 | | |
| Vascular invasion | 317 | | | | |
| No | 208 | Reference | | | |
| Yes | 110 | 1.344 (0.887–2.035) | 0.163 | | |
| POGK | 373 | | | | |
| Low | 187 | Reference | | | |
| High | 187 | 1.582 (1.112–2.249) | 0.011 * | 1.550 (0.973–2.471) | 0.065 |

*: $p < 0.05$, **: $p < 0.01$, ***: $p < 0.001$.

### 3.5. Diagnostic Value of POGK Gene Expression in HCC

ROC curve analysis revealed that the area under the receiver operating characteristic (ROC) curve (AUC) of POGK gene expression was 0.891, suggesting the high diagnostic value of this gene in HCC (Figure 5A). After stratifying according to patient characteristics, the AUC value of POGK gene expression was 0.881 for T1 and T2 stage (Figure 5B), 0.921 for T3 and T4 stage (Figure 5C), 0.901 for M0 (Figure 5D), 0.898 for N0 (Figure 5E), 0.878 for pathologic stage 1 and 2 (Figure 5F), 0.912 for pathologic stage 3 and 4 (Figure 5G), 0.864 for histologic grade 1 and 2 (Figure 5H), and 0.934 for histologic grade 3 and 4 (Figure 5I).

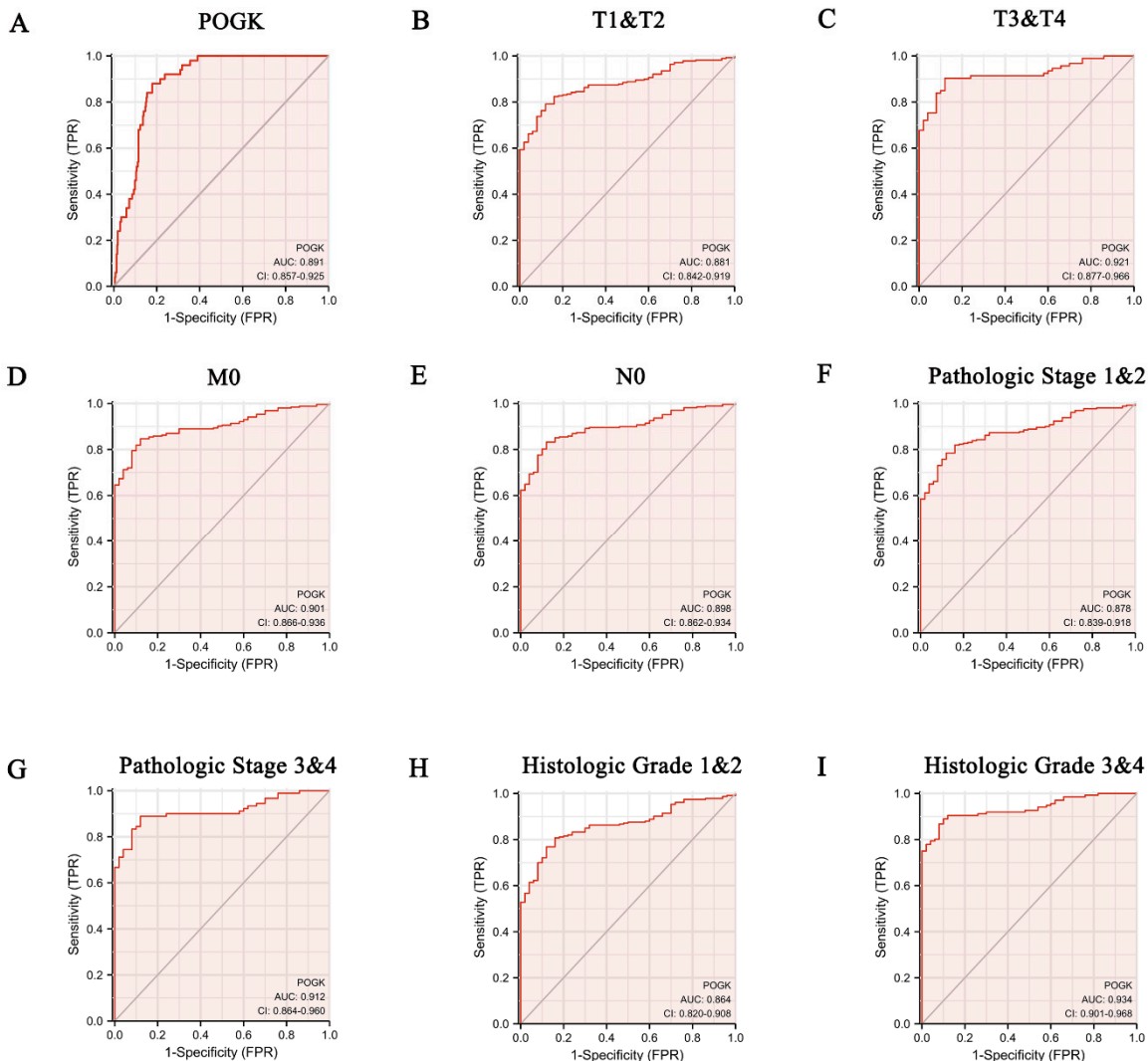

**Figure 5.** ROC curve for diagnostic value of POGK expression in patients with hepatocellular carcinoma. (**A**) in tumor tissue; (**B**) T1 and T2 stage; (**C**) T3 and T4 stage; (**D**) M0 stage; (**E**) N0 stage; (**F**) pathologic stage 1 and 2; (**G**) pathologic stage 3 and 4; (**H**) histologic grade 1 and 2; (**I**) histologic grade 3 and 4.

### 3.6. Functional Enrichment and Analyses of POGK Gene in HCC by GO Analysis

GO enrichment analysis was subsequently conducted to identify the biological processes, molecular functions, and cellular components related to the POGK gene. The POGK gene was significantly enriched in MFs including substrate-specific channel activity, ion channel activity, and metal ion transmembrane transporter activity. For CCs, significant enrichment in the apical plasma membrane, apical part of cells, and cell projection membrane was observed. The results are shown in Figure 6.

### 3.7. POGK-Related Signaling Pathways Identified by GSEA

GSEA showed a large difference in enriched MSigDB gene sets between the low and high POGK gene expression datasets. Based on the normalized enrichment score (NES), the most significantly enriched signaling pathways were then selected. Gene sets related to mitotic prometaphase, kinesins, homologous DNA pairing, and strand exchange, MET activates PTK2 signaling pathway, G1 to S cell cycle control, Aurora B pathway, ncRNAs involved in WNT signaling pathway, hepatitis C, and ncRNAs involved in the STAT3 signaling pathway, showed differential enrichment in the high POGK gene expression phenotype in HCC (Table 4; Figure 7A–I).

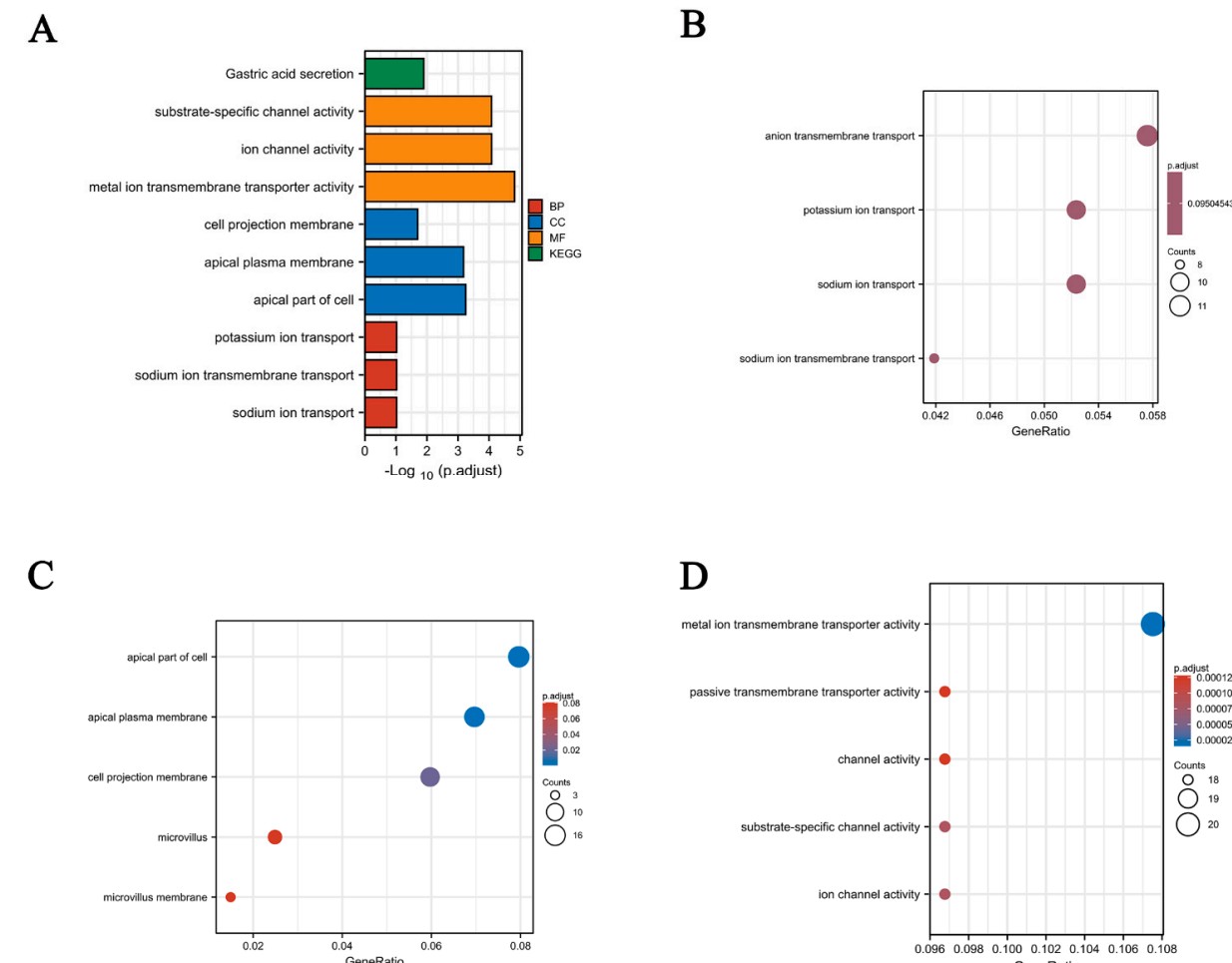

**Figure 6.** Significantly enriched GO annotations of the POGK gene in HCC. (**A**) GO functional enrichment analysis and KEGG pathway analysis; (**B**) biological process terms; (**C**) cell component terms and (**D**) molecular function terms. GO, Gene Ontology; KEGG, Kyoto Encyclopedia of Genes and Genomes; BP, biological process; CC, cellular component; MF, molecular function.

**Table 4.** Gene sets enriched in the high POGK expression phenotype.

| Gene Set Name | Size | ES | NES | *p*.adjust | *q* Values |
|---|---|---|---|---|---|
| Mitotic Prometaphase | 202 | 0.576702 | 2.79224 | 0.017522659 | 0.011278 |
| Kinesins | 61 | 0.671025 | 2.657603 | 0.017522659 | 0.011278 |
| Homologous DNA Pairing and Strand Exchange | 42 | 0.692287 | 2.568165 | 0.017522659 | 0.011278 |
| MET Activates PTK2 Signaling | 30 | 0.752825 | 2.545744 | 0.017522659 | 0.011278 |
| G1 to S Cell Cycle Control | 64 | 0.573636 | 2.289791 | 0.017522659 | 0.011278 |
| Aurora B Pathway | 39 | 0.623436 | 2.282505 | 0.017522659 | 0.011278 |
| ncRNAs Involved in WNT Signaling in Hepatocellular Carcinoma | 86 | 0.474281 | 2.018823 | 0.017522659 | 0.011278 |
| Hepatitis C and Hepatocellular Carcinoma | 50 | 0.506931 | 1.928244 | 0.019218103 | 0.012369 |
| ncRNAs Involved in STAT3 Signaling in Hepatocellular Carcinoma | 17 | 0.644878 | 1.902398 | 0.017522659 | 0.011278 |

Abbreviations: ES, enrichment score; NES, normalized enrichment score; Gene sets with *p*-value < 0.05 and *q*-value < 0.05 were considered as significantly enriched.

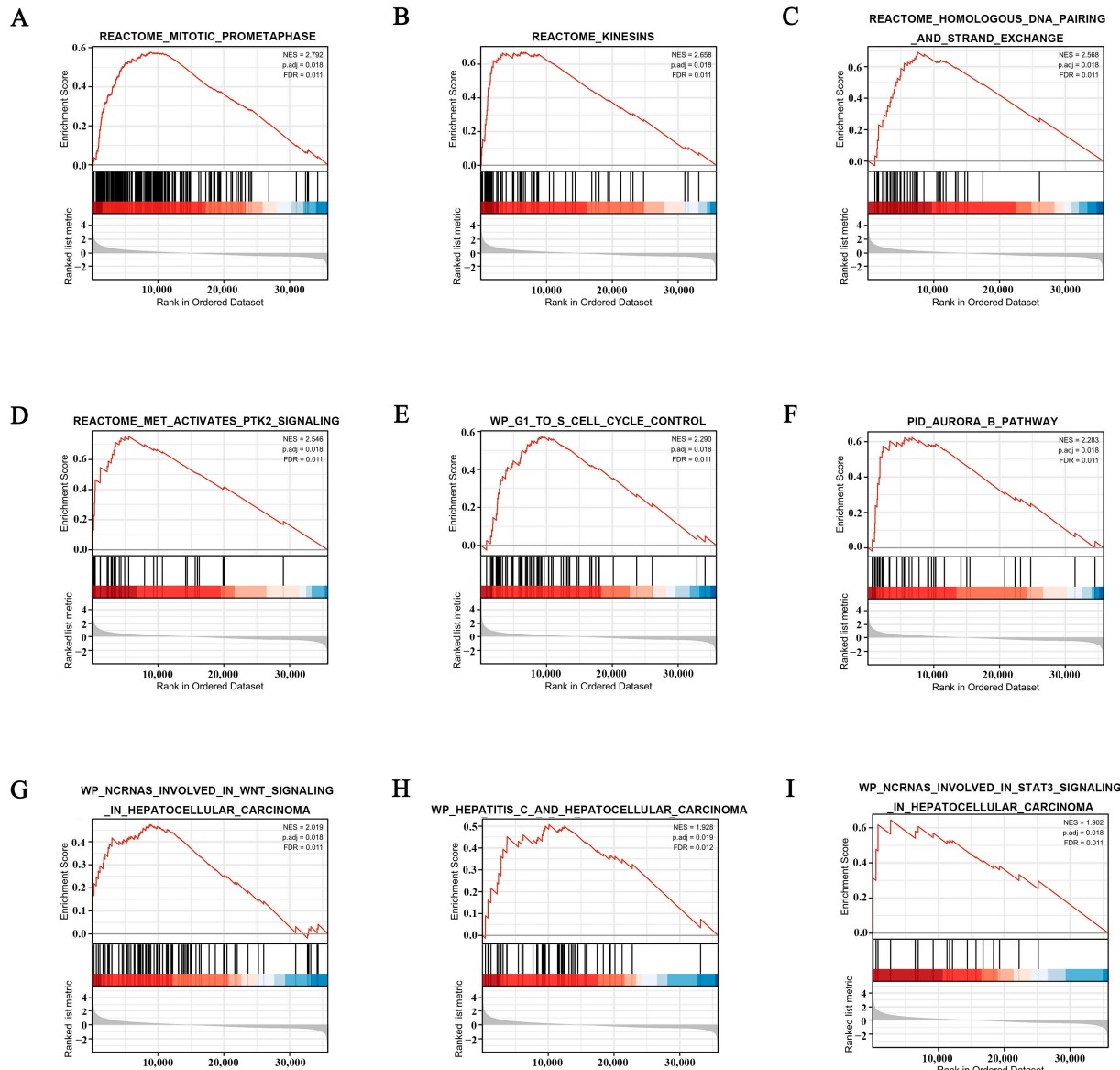

**Figure 7.** Enrichment plots from gene set enrichment analysis (GSEA) in patients with hepatocellular carcinoma with high POGK expression. (**A**) mitotic prometaphase; (**B**) kinesins; (**C**) homologous DNA pairing and strand exchange; (**D**) MET activates PTK2 signaling pathway; (**E**) G1 to S cell cycle control, (**F**) Aurora B pathway; (**G**) ncRNAs involved in WNT signaling pathway; (**H**) hepatitis C; (**I**) ncRNAs involved in the STAT3 signaling pathway.

### 3.8. Correlation between POGK Expression and Immune Infiltration

A negative correlation was found between POGK expression and the abundance of gamma delta T cells (Tgd), T cells, natural killer (NK) CD56 dim cells, CD8+ T cells, neutrophils, regulatory T cells (TReg), plasmacytoid pre-dendritic cells (pDC), dendritic cells (DC), cytotoxic cells, while a positive correlation was found with the abundance of helper T2 (Th2) cells, T helper cells, NK CD56 bright cells, central memory T cells (Tcm), follicular helper T cells (TFH) in the HCC microenvironment ($p_s < 0.05$) (Table 5 and Figure 8A–G).

**Table 5.** Correlation analysis between POGK and immune cells in GEO dataset.

| Cells | Coefficient of Correlation (Pearson) | *p* Value (Pearson) | Coefficient of Correlation (Spearman) | *p* Value (Spearman) |
|---|---|---|---|---|
| aDC | 0.080 | 0.122 | 0.056 | 0.279 |
| B cells | −0.060 | 0.247 | −0.081 | 0.119 |
| CD8 T cells | −0.167 | 0.001 ** | −0.180 | <0.001 *** |
| Cytotoxic cells | −0.399 | <0.001 *** | −0.427 | <0.001 *** |
| DC | −0.319 | <0.001 *** | −0.356 | <0.001 *** |
| Eosinophils | 0.071 | 0.169 | 0.070 | 0.176 |
| iDC | −0.050 | 0.334 | −0.085 | 0.100 |
| Macrophages | 0.088 | 0.090 | 0.059 | 0.251 |
| Mast cells | −0.087 | 0.094 | −0.057 | 0.269 |
| Neutrophils | −0.202 | <0.001 *** | −0.221 | <0.001 *** |
| NK CD56 bright cells | 0.134 | 0.010 * | 0.154 | 0.003 ** |
| NK CD56 dim cells | −0.118 | 0.022 * | −0.158 | 0.002 ** |
| NK cells | 0.041 | 0.425 | −0.026 | 0.617 |
| pDC | −0.309 | <0.001 *** | −0.310 | <0.001 *** |
| T cells | −0.121 | 0.019 * | −0.144 | 0.005 ** |
| T helper cells | 0.283 | <0.001 *** | 0.281 | <0.001 *** |
| Tcm | 0.161 | 0.002 ** | 0.148 | 0.004 ** |
| Tem | 0.082 | 0.112 | 0.053 | 0.311 |
| TFH | 0.134 | 0.010 * | 0.123 | 0.017 * |
| Tgd | −0.058 | 0.262 | −0.143 | 0.006 ** |
| Th1 cells | −0.042 | 0.419 | −0.048 | 0.351 |
| Th17 cells | −0.027 | 0.599 | −0.022 | 0.668 |
| Th2 cells | 0.386 | <0.001 *** | 0.369 | <0.001 *** |
| TReg | −0.227 | <0.001 *** | −0.235 | <0.001 *** |

Th: T helper cell; Tfh: Follicular helper T cell; Treg, regulatory T cell. *: $p < 0.05$, **: $p < 0.01$, ***: $p < 0.001$.

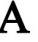

**A**

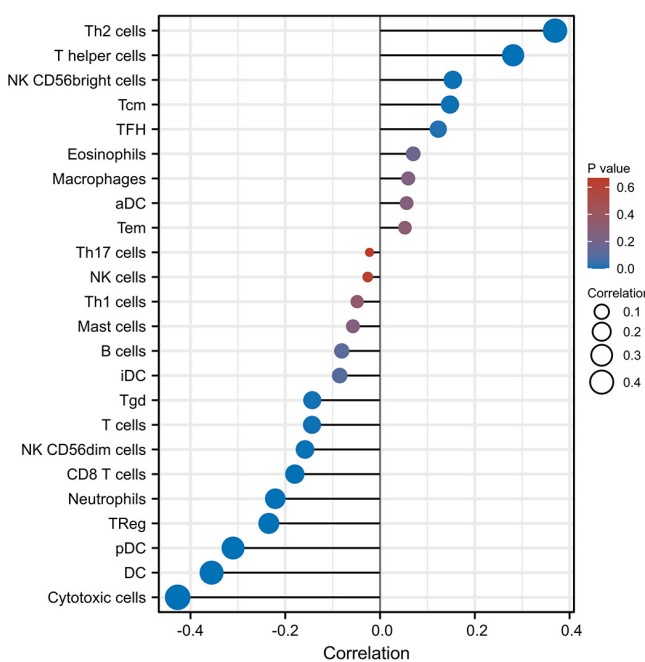

**Figure 8.** *Cont.*

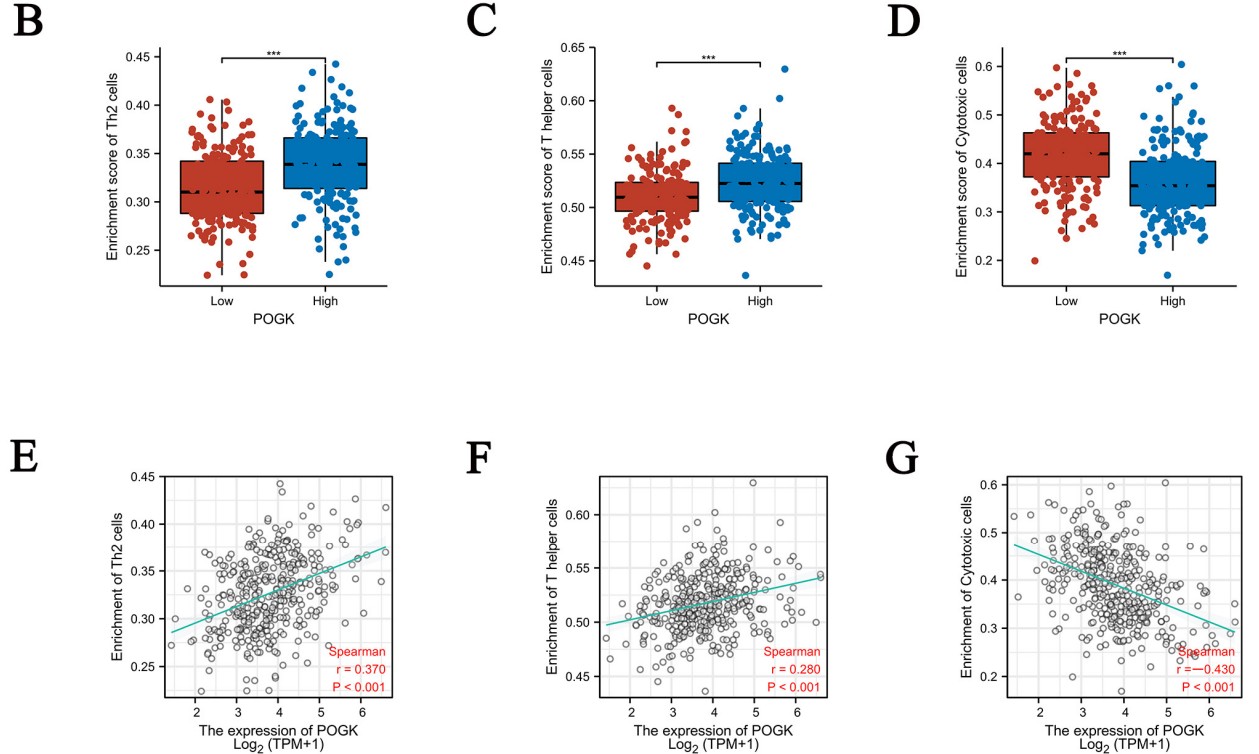

**Figure 8.** Correlation between POGK expression and immune cells in the tumor microenvironment. (**A**) Correlation between the relative abundances of 24 immune cells and POGK expression level. The absolute value of Spearman R was shown by the size of the dots. (**B–G**) Scatter plots and correlation charts showing the difference in Th2 cells, T helper cells, and cytotoxic cell enrichment levels between high and low POGK expression level groups. ***: $p < 0.001$.

## 4. Discussion

To the best of our knowledge, this is the first study to assess POGK gene expression and its potential prognostic impact on HCC. Herein, we found that POGK was upregulated in HCC patients and correlated with a poor prognosis. Furthermore, high POGK expression was involved in many signal pathways in HCC and correlated with the abundance of adaptive and innate immunocytes.

The past decade has witnessed unprecedented scientific advances, which have led to the discovery of many factors involved in HCC development and progression. In this study, POGK was significantly upregulated, and high POGK expression was associated with poor prognosis in HCC patients. Meanwhile, ROC curve analysis demonstrated the high diagnostic value of this gene in differentiating between HCC patients and healthy subjects (AUC = 0.891). It has been established that POGK contains a transposase domain at the C-terminus and a KRAB domain at the N-terminus. There is ample literature suggesting that KRAB proteins, including KRAB-ZFPs and KAP1, play important roles in neoplastic transformation [12–15]. A cluster of 16 KRAB-ZNFs was documented to be commonly upregulated across multiple cancer cohorts in a TCGA pan-cancer expression analysis [16]. Moreover, a systemic review reported that KRAB-ZFPs play oncogenic and suppressive roles in various cancers [17]. POGK, derived from the KRAB domain, may have similar roles to KRAB-ZFPs in neoplastic transformation. Moreover, pogo transposases are one of the superfamilies of IS630-Tc1-mariner (ITm), which represent the most prevalent DNA transposable elements (TEs) [18,19]. The evolution and diversity of pogo transposases has been widely documented in the literature [20,21]. The Pogo superfamily is widely distributed in animals and fungi, and has been reported in vertebrates with 12 genes, including POGK, pogo transposable element derived with ZNF domain (POGZ), Jrk helix-turn-helix protein (JRK), JRK-like (JRKL), centromere-associated protein B

(CENPB), CENPB DNA-binding domain containing 1 (CENPBD1), and Tigger transposable element-derived 2 to 7 (TIGD2-7) [22]. Overwhelming evidence substantiates that genes of the pogo superfamily are associated with certain malignancies. For instance, high expression of POGZ is reportedly associated with a poor prognosis of osteosarcoma [23], while JRK expression was aberrantly elevated in colorectal, breast, and ovarian cancers [24]. In addition, JRK expression predicts worse survival in soft tissue sarcomas [25]. For CENPB and CENPBD1, CENPB can be a serum biomarker for the diagnosis of lung cancer [26], while mRNA expression of CENPBD1 has prognostic value for survival in radio(chemo)therapy-treated head and neck squamous cell carcinoma [27]. POGK is a gene of the pogo superfamily with similar functions to other genes in the family and is widely thought to be involved in tumorigenesis, growth, and metastasis of HCC.

The function of POGK has been largely understudied. In this study, GO enrichment analysis showed that the POGK gene was significantly enriched in many BPs, CCs (such as apical plasma membrane, apical part of the cell, and cell projection membrane), and MFs (such as substrate-specific channel activity, ion channel activity, and metal ion transmembrane transporter activity). GSEA showed significant enrichment in pathways such as mitotic prometaphase, kinesins, homologous DNA pairing, and strand exchange; MET activates the PTK2 signaling pathway, G1 to S cell cycle control, Aurora B pathway, ncRNAs involved in WNT signaling pathway, hepatitis C, and ncRNAs involved in STAT3 signaling pathway) in the high POGK gene expression phenotype in HCC in this study. Meanwhile, POGK expression correlated with the abundance of immune cells in the tumor microenvironment of HCC. It has been shown that Pogo transposase contains a putative helix-turn-helix DNA binding domain indicating that it is a DNA transposon [28]. Current evidence suggests that KRAB proteins KRAB-ZFPs belong to the largest family of transcriptional regulators in higher vertebrates and mediate various processes related to development and physiology, such as heterochromatin induction in early development and TEs control, cell differentiation, and cellular metabolism [29]. Moreover, KRAB proteins have been demonstrated to control adaptive immune cell differentiation and function in mice and humans [9], which indicates the function and immune infiltration associated with POGK.

It is widely acknowledged that standard-of-care treatment for HCC is mainly based on tumor status and liver function. Although surgery remains the mainstay of curative treatment, it is indicated in only selected patients [30]. Molecular targeted therapy and immunotherapy have made vast progress over the past few years. Immune checkpoint inhibitor-based combinations have huge prospects for application as the first-line therapy in unresectable HCC [31]. Unfortunately, only a minority of HCC patients benefit from molecular-targeted therapy and immunotherapy. Accordingly, more clinical trial data are needed to support specific treatment strategies. Herein, we substantiated that POGK gene expression correlated with the abundance of immune cells in the tumor microenvironment of HCC, suggesting it can be a potent therapeutic target for this patient population, emphasizing the need for more studies.

There were some limitations in this study. First, although the data were collected from multicenter studies in public databases, this study lacked uniform intervention measures and further details on the patients, given its retrospective nature. Indeed, further experiments are warranted to assess the expression differences of POGK between HCC, adjacent tissue, and normal tissue. Although matched HCC tissues and adjacent normal tissues from HCC patients were analyzed for POGK expression in this study, the robustness of our findings was affected to a certain extent by the heterogeneity of the study population obtained from the TCGA. Moreover, we did not explore the potential mechanism of POGK in HCC. Further studies are thus necessary to explore the underlying mechanisms via POGK-knockout and POGK-overexpressed models.

In summary, high expression of POGK has a high diagnostic value and correlates with a poor prognosis in HCC patients. Moreover, POGK expression is correlated with immune

infiltration in HCC. These findings suggest that POGK has huge prospects for application as a new biomarker for HCC.

## 5. Conclusions

In conclusion, our study pointed out that POGK has a high diagnostic value for hepatocellular carcinoma, and the high expression of POGK is closely related to the poor prognosis of HCC pa-tients. At the same time, our data explain the biological function of POGK and its relevance to other signaling pathways, and demonstrate that POGK expression is associated with immune infiltration in HCC. These findings reveal that POGK may be a potential new biomarker for HCC and provide a theoretical basis for the development of targeted drugs in HCC.

**Author Contributions:** W.X., Y.H., Y.M., Y.Z., Q.L., S.Z., L.P., Z.G., Y.L. and J.L. contributed to the concept and design of the study. W.X., Y.H. and J.L. had full access to all the data in the study and took responsibility for the integrity of the data and the accuracy of the data analysis. W.X., Y.H. and Y.M. participated in the acquisition, analysis, and interpretation of the data. W.X. drafted the manuscript. Y.H., Y.L. and J.L. revised the manuscript. W.X., Y.H. and Y.L. obtained funding and validated POGK expression in the clinical samples. Y.Z., Q.L., S.Z. and L.P. assisted in downloading and analyzing the data. J.L., Y.L. and Z.G. took responsibility for the supervision of the study. All authors have read and agreed to the published version of the manuscript.

**Funding:** This study was supported by grants from the Clinical Research Program of the Third Affil-iated Hospital of Sun Yat-sen University (No. QHJH201808 to W.X.), Guangzhou Science and Tech-nology Project (No. 202102080064 to W.X.), Guangdong Basic and Applied Basic Research Foundation (No. 2020A1515110907 to Y.L.), National Natural Science Foundation of China (No. 82204447 to Y.L., No. 82202417 to Y.H.), Natural Science Foundation of Guangdong Province (No. 2022A1515011056 to Y.L.), and the Postdoctoral Science Foundation of China (No. 2019TQ0381 to Y.H.).

**Institutional Review Board Statement:** This study conformed to the Ethical Guidelines of the 1975 Declaration of Helsinki. The study protocol was approved by the Ethics Committee of the Third Affiliated Hospital of Sun Yat-sen University (approval No. (2020)02-172-02).

**Informed Consent Statement:** Informed consent was obtained from all subjects involved in the study.

**Data Availability Statement:** All the data and material are presented in the study.

**Conflicts of Interest:** The authors declare no conflict of interest.

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
