# Peer review of "High Expression of POGK Predicts Poor Prognosis in Patients with Hepatocellular Carcinoma"

_curroncol, doi:10.3390/curroncol29110682_

Round 1
Reviewer 1 Report (Previous Reviewer 1)
The authors have successfully addressed the comments from this reviewer. Now it is recommended for acceptance by the journal.
Author Response
We are extremely grateful to the Reviewer’s comment. Thank you.

Reviewer 2 Report (Previous Reviewer 2)
All my questions and comments are addressed. It can be published.
Author Response
We are extremely grateful to the Reviewer’s comment. Thank you.

Reviewer 3 Report (New Reviewer)
The study assesses a current, timely topic in HCC.
We recommend some changes:
- We believe this article is suitable for publication in the journal although major revisions are needed. The main strengths of this paper are that it addresses an interesting and very timely question and provides a clear answer, with some limitations. Certainly, the study is limited to a population quite heterogeneous from the TCGA. Thus, the authors should better highlight the limitations of the current paper.
- A linguistic revision should be performed by a professional service.
- The background of the changing scenario of medical treatment in HCC should be better discussed, and some recent papers regarding this topic should be included (PMID: 34431725).
Major changes are necessary.
Author Response
- We believe this article is suitable for publication in the journal although major revisions are needed. The main strengths of this paper are that it addresses an interesting and very timely question and provides a clear answer, with some limitations. Certainly, the study is limited to a population quite heterogeneous from the TCGA. Thus, the authors should better highlight the limitations of the current paper.
Reply: We thank the Reviewer for these comments. As suggested, we added limitations to the discussion section of the revised manuscript.
- A linguistic revision should be performed by a professional service.
Reply: We concur with the Reviewer. The revised manuscript has been sent to an English native speaker for proofreading.
- The background of the changing scenario of medical treatment in HCC should be better discussed, and some recent papers regarding this topic should be included (PMID: 34431725).
Reply: We are extremely grateful to the Reviewer for this comment. Corresponding changes have been made to the discussion section of the revised manuscript, as recommended.

Round 2
Reviewer 3 Report (New Reviewer)
acceptance
This manuscript is a resubmission of an earlier submission. The following is a list of the peer review reports and author responses from that submission.
Round 1
Reviewer 1 Report
The manuscript entitled "High expression of POGK predicts poor prognosis in patients 2 with hepatocellular carcinoma" reports bioinformatics findings from analyzing public databases for the POGK gene. Although it lacks wet-lab results, the findings may provide useful for the field to use CCGs as potential markers or drug targets.
The manuscript was well-prepared and does not need major language revisions. However, the authors are strongly recommended to read it several times more to make sure no errors will appear eventually.
Before this manuscript can be recommended by this reviewer for acceptance, the authors will need to address at least one of the following points:
1. Do qPCR, Western blot, or staining using tissue samples from human patients to validate their findings of higher POGK expression.
2. Validate the diagnostic value of POGK expression using a new set of patients.
3. Correlate POGK expression with immune cells using patient samples or in vitro model systems.
4. Do functional study to show the role of unregulated expression of POGK in HCC; cell lines are acceptable.
Author Response
We are extremely grateful to the Reviewer’s comment. Tissue microarrays (TMAs) and Immunohistochemical (IHC) staining were used to analyse POGK expression from clinical samples of 30 HCC patients. The results are shown in new Figure 2.

Reviewer 2 Report
The authors present a in silico study of POGK in hepatocellular carcinoma (HCC). The language is good and I did not find grammatical errors. The prognostic value of POGK in HCC is not new and has been demonstrated in the study conducted by The Human Protein Atlas (https://www.proteinatlas.org/ENSG00000143157-POGK/pathology/liver+cancer). However, the paper provides additional correlations between POGK expression and clinicopathologic parameters. Also, the paper studied potential pathways regulated by POGK and therefore provides a mechanistic explanation of POGK's involvement in prognosis. Overall, I think it can be accepted if the following issues can be clarified:
1. Page 1 line 35: Instead of "predictive", I feel "prognostic" might be more appropriate since the paper demonstrate the patients with POGK-high HCC have a poor prognosis.
2. Page 4 figure 1C: What is LIHC and why is it highlighted?
3. Tables: may consider highlighting those characteristics reaching significant values.
4. Table 2: confusing. I am not sure Logistics regression based on POGK expression can be used to claim POGK is associated with poor prognosis (page 6, line 146 to 148). It can only demonstrate that how much the clinical characteristics in the table are related to (or predictive of) POGK gene expression. This is exactly the authors want to say because the table is under "3.3 Correlation between POGK expression and clinical characteristics“. It can be corrected by saying "POGK expression was associated with clinicopathologic characteristics typically associated with tumor aggressiveness".
5. There are conflicting data between table 1 and figure 3 (by the way, texts in figure 3 are very small and difficult to read). In table 1, the p values for OS, DSS, and PFI are not significant. However, in figure 3, OS and DSS have significant p values.
6. It appears POGK expression is not an independent risk factor because it failed to reach a significant value in multivariate cox regression as the authors stated. If that is the case, why is the conclusion "3.4. High expression of POGK is an independent risk factor for survival in HCC"?
7. The ROC is also confusing too. The sensitivity and specificity are based on how the differential expression of POGK in tumor versus normal liver can be used to diagnose HCC. It does not consider the expression of POGK in abnormal liver or mimickers of HCC. I am not sure POGK can be used to diagnose HCC since it is upregulated in many other cancers.
Author Response
- Page 1 line 35: Instead of "predictive", I feel "prognostic" might be more appropriate since the paper demonstrate the patients with POGK-high HCC have a poor prognosis.
Reply: We are extremely grateful to the Reviewer for pointing this out. It has been corrected.
- Page 4 figure 1C: What is LIHC and why is it highlighted?
Reply: LIHC is the abbreviation of “liver hepatocellular carcinoma”. Since our study aims to investigate the diagnostic and prognostic value of high POGK expression in hepatocellular carcinoma, LIHC is highlighted.
- Tables: may consider highlighting those characteristics reaching significant values.
Reply: We concur with the Reviewer. All p < 0.05 are highlighted in Table 1, 2, 3 and 5.
- Table 2: confusing. I am not sure Logistics regression based on POGK expression can be used to claim POGK is associated with poor prognosis (page 6, line 146 to 148). It can only demonstrate that how much the clinical characteristics in the table are related to (or predictive of) POGK gene expression. This is exactly the authors want to say because the table is under "3.3 Correlation between POGK expression and clinical characteristics“. It can be corrected by saying "POGK expression was associated with clinicopathologic characteristics typically associated with tumor aggressiveness".
Reply: We are extremely grateful to the Reviewer for pointing this out. It has been corrected.
- There are conflicting data between table 1 and figure 3 (by the way, texts in figure 3 are very small and difficult to read). In table 1, the p values for OS, DSS, and PFI are not significant. However, in figure 3, OS and DSS have significant p values.
Reply: We are extremely grateful to the Reviewer for pointing this out. The data of Table 1 is retrieved from the TCGA database, and the data in Figure 3 is retrieved from the website Kaplan Meier Plotter (doi: 10.1098/rsos.181006). Therefore, there will be inconsistencies after processing different data sets of two databases. In Table 1, chi square test was carried out using TCGA data, and the overall survival rate of the two groups was found to be of no survival significance. Figure 3 uses a professional evaluation method (kaplan meier), which we believe is reliable for evaluating the survival data of patients.
- It appears POGK expression is not an independent risk factor because it failed to reach a significant value in multivariate cox regression as the authors stated. If that is the case, why is the conclusion "3.4. High expression of POGK is anindependent risk factor for survival in HCC"?
Reply: We are extremely grateful to the Reviewer for pointing this out. It has been corrected. After statistical analysis, we believe that POGK can predict the survival and prognosis of HCC. We have deleted the word “independent” in the text.
- The ROC is also confusing too. The sensitivity and specificity are based on how the differential expression of POGK in tumor versus normal liver can be used to diagnose HCC. It does not consider the expression of POGK in abnormal liver or mimickers of HCC. I am not sure POGK can be used to diagnose HCC since it is upregulated in many other cancers.
Reply: We concur with the Reviewer. This study mainly investigated the expression of POGK in normal liver tissues and HCC tissues, and did not consider the expression of POGK in the abnormal liver or mimics of HCC. Therefore, the research results obtained in this paper are considered to be obtained in a group composed of normal people and liver cancer patients, and the specific efficacy of POGK in diagnosing HCC is discussed. We will elaborate and correct in the results and conclusions. Thank you for your comments!
